# The Emerging Role of SIRT7 in Glucose and Lipid Metabolism

**DOI:** 10.3390/cells13010048

**Published:** 2023-12-25

**Authors:** Kazuya Yamagata, Tomoya Mizumoto, Tatsuya Yoshizawa

**Affiliations:** 1Department of Medical Biochemistry, Faculty of Life Sciences, Kumamoto University, Kumamoto 860-8556, Japan; waterbook50@gmail.com (T.M.); yoshizaw@kumamoto-u.ac.jp (T.Y.); 2Center for Metabolic Regulation of Healthy Aging, Faculty of Life Sciences, Kumamoto University, Kumamoto 860-8556, Japan

**Keywords:** sirtuin, SIRT7, SIRT1, SIRT6, diabetes, obesity

## Abstract

Sirtuins (SIRT1–7 in mammals) are a family of NAD+-dependent lysine deacetylases and deacylases that regulate diverse biological processes, including metabolism, stress responses, and aging. SIRT7 is the least well-studied member of the sirtuins, but accumulating evidence has shown that SIRT7 plays critical roles in the regulation of glucose and lipid metabolism by modulating many target proteins in white adipose tissue, brown adipose tissue, and liver tissue. This review focuses on the emerging roles of SIRT7 in glucose and lipid metabolism in comparison with SIRT1 and SIRT6. We also discuss the possible implications of SIRT7 inhibition in the treatment of metabolic diseases such as type 2 diabetes and obesity.

## 1. Introduction

Sirtuins (SIRT1–7 in mammals) are evolutionally conserved nicotinamide adenine dinucleotide (NAD+)-dependent deacetylases and deacylases that regulate a wide variety of biological processes, including metabolism, stress responses, genomic stability, tumorigenesis, and aging. The seven SIRTs share a highly conserved NAD+-binding and catalytic core domain, while the N- and C- terminal regions vary structurally and contribute to differences in tissue expression, subcellular localization, and enzymatic activities [1,2,3]. SIRT1 and SIRT6 are located predominantly in the nucleus (nuclear sirtuin). SIRT7 is another nuclear sirtuin and is present in the nucleolus and nucleoplasm but also resides in the cytoplasm. SIRT2 is cytoplasmically localized, whereas SIRT3, SIRT4, and SIRT5 are located mainly in the mitochondria [4]. Regarding enzymatic activities, SIRT1, SIRT2, and SIRT3 have strong deacetylase activity, while SIRT4, SIRT5, and SIRT6 have either weak or undetectable deacetylase activity in various in vitro assays. SIRT4 has ADP-ribosyltransferase activity [5], and SIRT5 has demalonylase and desuccinylase activities [6]. SIRT6 preferentially catalyzes deacylation of long-chain fatty-acyl groups from target proteins [7,8].

SIRT7 is ubiquitously expressed in almost all tissues, with the exception of skeletal muscle [9]. The enzymatic activity of SIRT7 was unknown for a long time, but Baber et al. reported that SIRT7 possesses histone H3K18 selective deacetylation activity [10]. Since this breakthrough, a number of target proteins and physiological functions of SIRT7 have been identified (Table 1) [10,11,12,13,14,15,16,17,18,19,20,21,22,23,24,25,26,27,28,29,30,31,32,33,34,35,36,37,38,39,40,41,42,43,44] and it is now clear that SIRT7 plays critical roles in many biological processes, including gene expression, genome stability, stress response, and metabolism. Recent studies have also revealed that in addition to deacetylase activity, SIRT7 has both deacylase and ADP-ribosyltransferase activities [39,40,41,42,43,44]. SIRT1 and SIRT6 are crucial regulators of fat and glucose metabolism, but the metabolic roles of SIRT7 received little research attention until recently [2,45,46,47]. However, accumulating evidence indicates that SIRT7 also plays a critical role in the regulation of metabolism. This review focuses on the recent progress in elucidating the role of SIRT7 in lipid and glucose metabolism in comparison with SIRT1 and SIRT6. Other cellular and biological roles of SIRT7, such as stress response, are covered in several recent reviews [48,49,50,51]. A better understanding of the metabolic roles of SIRT7 might be useful in the development of new strategies for treating metabolic diseases such as obesity and type 2 diabetes.

## 2. The Role of Nuclear Sirtuins in Lipid Metabolism

White adipose tissue (WAT) is the predominant type of fat in humans, and white adipocytes serve as a storage depot for excess energy as triacylglycerol. During fasting, triacylglycerol stored in adipocytes is hydrolyzed to produce fatty acids. These fatty acids are then oxidized in the liver, muscles, and so on to generate ATP. Fatty acids are also used for heat production in brown adipose tissue (BAT). Recent studies have clarified that SIRT7 influences diverse aspects of lipid homeostasis in these organs.

### 2.1. Adipogenesis

Adipogenesis is a well-orchestrated multistage differentiation process in which fibroblast-like preadipocytes mature into lipid-laden, insulin-responsive adipocytes. This process is regulated by a cascade of transcriptional factors, among which nuclear receptor peroxisome proliferator-activated γ (PPARγ) functions as a master regulator [52,53,54]. SIRT1 represses the activity of PPARγ by promoting the docking of co-repressors such as nuclear receptor co-repressor (NCoR) and silencing mediator of retinoid and thyroid hormone receptors (SMRT) on the promoters of PPARγ target genes [55]. Thus, SIRT1 acts as a negative regulator of adipogenesis.

Mitotic clonal expansion is an event that occurs early in the process of adipogenesis [52,53]. Kinesin family member 5C (KIF5C) is a negative regulator of mitotic clonal expansion, and SIRT6 negatively regulates KIF5C expression through deacetylation of H3K9 and H3K56 at the promoter. Thus, SIRT6 is essential for adipogenesis because it regulates mitotic clonal expansion [56]. However, it has also been reported that SIRT6 inhibits preadipocyte differentiation by activating the AMP-activated protein kinase (AMPK) pathway [57].

SIRT7 is also implicated in the regulation of adipogenesis [58,59]. Mechanistically, SIRT7 restricts SIRT1 activity by inhibiting SIRT1 autodeacetylation activity [59]. As described above, SIRT1 suppresses adipogenesis [55]. Thus, SIRT7 promotes adipogenesis by inhibiting SIRT1 activity and enhancing PPARγ activity (Figure 1A). These findings clearly indicate that SIRT7 and SIRT1 play opposite roles in adipogenesis.

### 2.2. Lipolysis and Lipogenesis

White adipocytes are the main storage compartment of triacylglycerol in the body, and the storage mechanism is regulated by the balance of lipid synthesis and hydrolyzation (lipolysis). In the fasting state, the rate of lipolysis increases, and free fatty acids (FFAs) are released into the blood, while the outflow of FFAs from adipocytes decreases in the fed state. In addition to its role in adipogenesis, PPARγ also plays a crucial role in increasing the uptake and trapping of FFAs in mature adipocytes by regulating the expression of lipoprotein lipase (LPL), fatty acid transport protein (FATP), and CD36 [54,60]. SIRT1 represses PPARγ activity as described above and enhances lipolysis in differentiated mature adipocytes [55]. Consistently, the release of FFAs from white adipocytes upon fasting was compromised in *Sirt1*^+/−^ mice [55].

Adipose triglyceride lipase (ATGL) is a rate-limiting enzyme that catalyzes the initial step of lipolysis and converts triacylglycerol to diacylglycerol [61,62]. The expression of ATGL is controlled by Forkhead Box O1 (FOXO1) in adipocytes, and SIRT6 deficiency represses the transcription of *Atgl* by increasing the acetylation of FOXO1, thereby promoting its nuclear exclusion [63,64]. Therefore, both SIRT1 and SIRT6 promote lipolysis in white adipocytes.

PPARγ exists as two isoforms, PPARγ1 and PPARγ2, as a result of alternative splicing and different promoter usage [54]. PPARγ1 is expressed in many tissues, but the expression of PPARγ2, which contains 30 additional amino acids at the N-terminus, is restricted to adipose tissue. SIRT7 binds to PPARγ2 and deacetylates it at K382 [20]. C3H10T1/2-derived adipocytes expressing PPARγ2^K382Q^ (a mimic of acetylated K) accumulate much less fat compared with adipocytes expressing PPARγ2^K382R^ (a mimic of deacetylated K).

Intriguingly, the expression of lipogenic genes, including *Fasn* (encoding fatty acid synthase), *Acaca* (encoding acetyl CoA carboxylase α), *Scd1* (encoding stearoyl CoA desaturase), and *Srebp1c* (encoding sterol regulatory element binding protein 1c), were selectively decreased in both the PPARγ2^K382Q^ cells and WAT of *Sirt7* knockout (KO) mice. These findings indicate that SIRT7-dependent PPARγ2 deacetylation at K382 preferentially promotes lipogenesis in white adipocytes (Figure 1B) [20]. SIRT7 is also reported to promote lipogenesis by inducing the methylation of SREBP1a in tumors [40].

### 2.3. Thermogenesis

In addition to WAT, mammals also possess BAT, which plays a crucial role in whole-body energy expenditure through non-shivering thermogenesis. Uncoupling protein 1 (UCP1) is a brown adipocyte-specific protein located within mitochondria that catalyzes a protein leak across the inner mitochondrial membrane. Brown adipocytes dissipate energy as heat through the activation of UCP1 by uncoupling fuel oxidation from ATP synthesis [65,66]. β3-adrenergic receptors are abundantly expressed in brown adipocytes, and β-adrenergic receptor stimulation (e.g., cold exposure) increases the generation of cAMP and leads to protein kinase A (PKA) activation. PKA-dependent phosphorylation of p38α map kinase phosphorylates activating transcription factor 2 (ATF2) to induce *Ppargc1a* (encoding peroxisome proliferator-activated receptor gamma coactivator 1alpha; PGC1α) transcription. PGC1α co-activates transcription factors (e.g., PPARγ) assembled on the *Ucp1* enhancer, thereby increasing *Ucp1* gene expression [67,68]. Transgenic mice overexpressing SIRT1 display enhanced BAT function with increased thermogenesis and energy expenditure [69], whereas SIRT1-deficient mice exhibit decreased thermogenesis and energy expenditure when fed a high-fat diet (HFD) [70]. Thus, SIRT1 functions as a positive regulator of BAT activity. Beige adipocytes are an inducible form of UCP1^+^ thermogenic adipocytes that sporadically reside within subcutaneous WAT [66]. SIRT1 also promotes the induction of beige adipocytes in WAT (i.e., browning) by deacetylating PPARγ on K268 and K293 [71]. SIRT1-dependent PPARγ deacetylation enhances the browning of WAT by regulating ligand-dependent coactivator/corepressor exchange at the PPARγ transcriptional complex.

SIRT6 also positively regulates thermogenesis in brown adipocytes and promotes the induction of beige adipocytes [72]. SIRT6 increases the transcription of *Ppargc1a* by enhancing the binding of phosphorylated ATF2 to its promoter region. Adipocyte-specific SIRT6-deficient mice exhibited impaired thermogenesis and reduced whole-body energy expenditure [72]. SIRT6 also regulates lipid catabolism and thermogenesis in BAT by promoting the deacetylation of FOXO1 [73].

In contrast to SIRT1 and SIRT6, SIRT7 suppresses thermogenesis and energy expenditure by regulating the function of brown adipocytes [21]. Whole-body and BAT-specific *Sirt7* KO mice showed higher body temperature and energy expenditure with increased UCP1 protein expression in BAT. Mechanistically, SIRT7 deacetylates insulin-like growth factor 2 mRNA-binding protein 2 (IMP2/IGF2BP2), an RNA-binding protein that inhibits the translation of *Ucp1* mRNA [74], thereby enhancing its inhibitory action on *Ucp1* (Figure 1C). SIRT7 deficiency also enhances the browning of inguinal WAT [21], although the detailed mechanisms are not known. These findings indicate that SIRT7 suppresses not only thermogenesis in BAT but also the induction of beige adipocytes.

### 2.4. Adipose Tissue Inflammation

Excess energy intake is associated with the accumulation of lipids in adipocytes and the expansion of adipose tissue. The accumulation of fat mass induces low-grade chronic inflammation by producing proinflammatory cytokines such as TNF-α, IL-6, and MCP-1 in adipose tissue, and obesity-induced inflammation induces insulin resistance in the liver and muscles [75,76]. Nuclear factor (NF)-κB is a central transcription factor that controls the expression of genes involved in inflammation, and SIRT1 represses the activity of NF-κB by deacetylating the p65 subunit of NF-κB [77]. Thus, SIRT1 exerts anti-inflammatory effects in adipose tissue by regulating proinflammatory transcription and also improves insulin sensitivity [78,79].

SIRT6 also has an anti-inflammatory role. SIRT6 is recruited to the promoters of NF-κB target genes through interactions with NF-κB and suppresses the expression of NF-κB target genes by deacetylating histone H3K9 at the target gene promoters [80]. Consistently, adipocyte-specific *Sirt6* KO mice exhibit increased insulin resistance and inflammation in the adipose tissue [64,81].

In contrast, SIRT7 deficiency ameliorates inflammatory responses such as induction of *Tnfa* and *Il6* both in vitro and in vivo [32,82,83,84]. SIRT7 interacts with a small GTPase, Ras-related nuclear antigen (Ran), and deacetylates Ran at K37. In the absence of SIRT7, acetylation of Ran facilitates the formation of the NF-κB export cargo complex, thereby enhancing the nuclear exclusion of NF-κB (Figure 1D) [32]. Indeed, the expression of genes involved in inflammation was decreased in the WAT of *Sirt7* KO mice [85]. Further studies are needed to elucidate the roles of SIRT7 in adipose tissue inflammation, but findings to date suggest that SIRT7 might control inflammation in WAT.

### 2.5. Hepatic Lipid Metabolism

The liver also plays an important role in the regulation of lipid metabolism. After uptake by the liver, FFAs, which are released from adipocytes, are β-oxidized (β-oxidization) to acetyl-CoA in order to produce ATP. In the fed state, fatty acids are synthesized from acetyl-CoA in the liver (lipogenesis). PPARα is a key transcription factor for the regulation of genes involved in fatty acid uptake and β-oxidation, and PGC1α stimulates the transcriptional activity of PPARα [86]. SIRT1 increases PPARα target genes, including *Mcadh* (encoding medium-chain acyl-CoA dehydrogenase), *Aox* (encoding acyl CoA oxidase), and *Ehhadh* (encoding enoyl-CoA hydratase/3-hydroxyacyl CoA dehydrogenase), by deacetylating and activating PGC1α [87]. Hepatic deletion of SIRT1 in mice resulted in hepatic lipid accumulation under HFD feeding [87]. SREBP1c is a key transcription factor that regulates the expression of lipogenic genes, including *Fasn*, *Acaca*, and *Scd1* in the liver. SIRT1 deacetylates and inhibits SREBP1c activity, and transgenic mice overexpressing SIRT1 are protected from HFD-induced hepatic steatosis [88,89]. Therefore, SIRT1 both promotes β-oxidation and inhibits fat synthesis in the liver.

SIRT6 also stimulates fatty acid β-oxidation and inhibits lipogenesis in the liver, and liver-specific SIRT6 KO mice exhibited hepatic lipid accumulation [90]. MicroRNA-122 (miR-122) is a key regulator of fatty acid metabolism, and the inhibition of hepatic miR-122 expression stimulates β-oxidation and reduces fatty acid synthesis [91]. SIRT6 decreases hepatic lipid accumulation by suppressing the expression of miR-122 through the deacetylation of H3K56 in the promoter region [92]. SIRT6 also inhibits hepatic lipogenesis by repressing *Srebp1* and its target genes [93].

In contrast to SIRT1 and SIRT6, SIRT7 functions as a positive regulator of hepatic lipid accumulation, and *Sirt7* KO mice are resistant to HFD-induced fatty liver [85]. The nuclear receptor TR4 stimulates lipid uptake, triacylglycerol synthesis, and storage in the liver by regulating the expression of *Cd36*, *Cidea* (encoding cell death inducing DNA fragmentation factor A (DFFA)-like effector a), *Cidec* (encoding cell death inducing DFFA like effector c), *Mogat1* (encoding monoacylglycerol O-acyltransferase 1), and *Pparg*. SIRT7 increases TR4 protein expression by deacetylating DDB1 and inhibiting the degradation of TR4 via the CUL4B/DDB1/DCAF1 E3 ubiquitin ligase complex, leading to hepatic lipid accumulation (Figure 1E) [19,85]. However, contradictory results have also been reported [94]. GA-binding protein (GABP) is a nuclear transcription factor that controls mitochondrial biogenesis and function. Because SIRT7 increases GABP activity by deacetylating GABPβ1, aged *Sirt7* KO mice are reported to exhibit hepatic microvesicular steatosis due to mitochondrial dysfunction (Figure 1F) [22]. However, the expression of several nuclear-encoded mitochondrial transcripts regulated by GABP was not altered in the liver of young *Sirt7* KO mice [21]. Therefore, SIRT7 has different roles in hepatic lipid homeostasis depending on the conditions.

## 3. Nuclear Sirtuins in Glucose Metabolism

Glucose is a major fuel for most tissues. In the fed state, glucose is metabolized to pyruvate by glycolysis. In aerobic tissues, pyruvate is further metabolized to acetyl-CoA, which can enter the citric acid cycle for complete oxidation; this is linked to the formation of ATP in the process of oxidative phosphorylation. In the fasting state, glucose is produced from noncarbohydrate precursors, including amino acids, lactate, and glycerol, in the process of gluconeogenesis. Along with SIRT1 and SIRT6, SIRT7 is also involved in the regulation of these metabolic pathways.

### 3.1. Gluconeogenesis

The liver is the main site of gluconeogenic tissue. SIRT1 enhances gluconeogenesis by increasing transcription of the gluconeogenic genes *G6pc* (encoding glucose-6-phosphatase) and *Pck1* (encoding phosphoenolpyruvate carboxykinase 1) in the liver through the deacetylation of PGC1α and FOXO1 [95,96,97]. However, SIRT1 also inhibits gluconeogenesis by promoting the ubiquitin-mediated degradation of CREB-regulated transcription coactivator 2 (CRTC2), a coactivator of gluconeogenesis [98]. Therefore, SIRT1 has a dual role in the control of gluconeogenesis.

SIRT6 suppresses gluconeogenesis by deacetylating general control non-repressed protein 5 (GCN5), which then increases PGC1α acetylation and suppresses its stimulation of gluconeogenesis [99]. SIRT6 also downregulates the transcription of *G6pc* and *Pck1* through the deacetylation of FOXO1 [100]. However, it has also been reported that liver-specific SIRT6 transgenic mice and control mice showed similar blood glucose levels [101].

SIRT7 also plays a critical role in the regulation of gluconeogenesis [23,85]. Circadian protein cryptochrome 1 (CRY1) represses hepatic gluconeogenesis by promoting FOXO1 degradation [102,103]. SIRT7 destabilizes CRY1 protein expression through its deacetylation, thereby inhibiting the CRY1-mediated suppression of gluconeogenesis (Figure 2A) [23]. SIRT7 is also reported to increase the transcription of *G6pc* through the deacetylation of H3K18 in the *G6pc* promoter (Figure 2B) [104].

### 3.2. Glycolysis and Mitochondrial Function

Glycolysis is a major route for glucose metabolism, in which glucose is catabolized to pyruvate. When oxygen is in short supply, pyruvate is the end-product of glycolysis. Under aerobic conditions, pyruvate is further metabolized in the mitochondria. SIRT1 inhibits glycolysis by activating PGC1α, which suppresses glycolytic genes in the fasting state [95]. SIRT1 also suppresses glycolysis through the deacetylation of phosphoglycerate mutase 1 [105]. Hypoxia inducible factor 1 (HIF1) is a central transcription factor in the cellular response to oxygen and nutrient stress. HIF1 enhances glycolytic flux by upregulating the expression of glycolytic genes and inhibits mitochondrial oxidation of pyruvate via the stimulation of *Pdk* (encoding pyruvate dehydrogenase kinase) genes [106]. SIRT1 also inhibits glycolysis by suppressing the activity of HIF1 [107].

SIRT6 functions as a corepressor of HIF1 and suppresses the expression of multiple glycolytic genes [108]. Consequently, SIRT6 deficiency increases glucose uptake with upregulation of glycolytic enzymes, causing severe hypoglycemia in mice [108].

SIRT7 is also implicated in the regulation of glycolysis. SIRT7 overexpression decreases HIF1 protein expression levels, and the effect is independent of the enzymatic activity of SIRT7 (Figure 2C) [109]. In addition, SIRT7 is reported to reduce the activity of phosphoglycerate kinase 1 (PGK1) through its deacetylation in cancer cells (Figure 2D) [25]. Although the roles of SIRT7 in glycolysis are not directly explored in vivo, these observations suggest that SIRT7 contributes to the inhibition of glycolysis. Furthermore, SIRT7 modulates mitochondrial biogenesis and function. As described above, SIRT7 functions as a crucial regulator of mitochondria in hepatocytes by activating GABP through the deacetylation of GABPβ1 [22]. In high-glucose conditions, SIRT7 is methylated at arginine 388 (R388), which inhibits the H3K18-deacetylase activity of SIRT7 and stimulates mitochondrial biogenesis [110]. SIRT7 also suppresses mitochondrial biogenesis and respiration by interacting with nuclear respiratory factor 1 (NRF1) in hematopoietic stem cells [37].


Figure 2Roles of SIRT7 in glucose metabolism. (**A**) SIRT7 stimulates gluconeogenesis through the destabilization of CRY1 [23]. (**B**) SIRT7 increases transcription of the gluconeogenic gene *G6pc* through the deacetylation of histone H3K18 [104]. (**C**) SIRT7 inhibits glycolysis by decreasing the HIF1 protein expression levels [109]. (**D**) SIRT7 also suppresses glycolysis through the deacetylation of PGK1 [25]. (**E**) SIRT7 deficiency increases energy expenditure by increasing the expression levels of the batokines *Fgf21* and *Nrg4* in BAT [21]. (**F**) SIRT7 suppresses AKT activation by deacetylating FKBP51 [26].
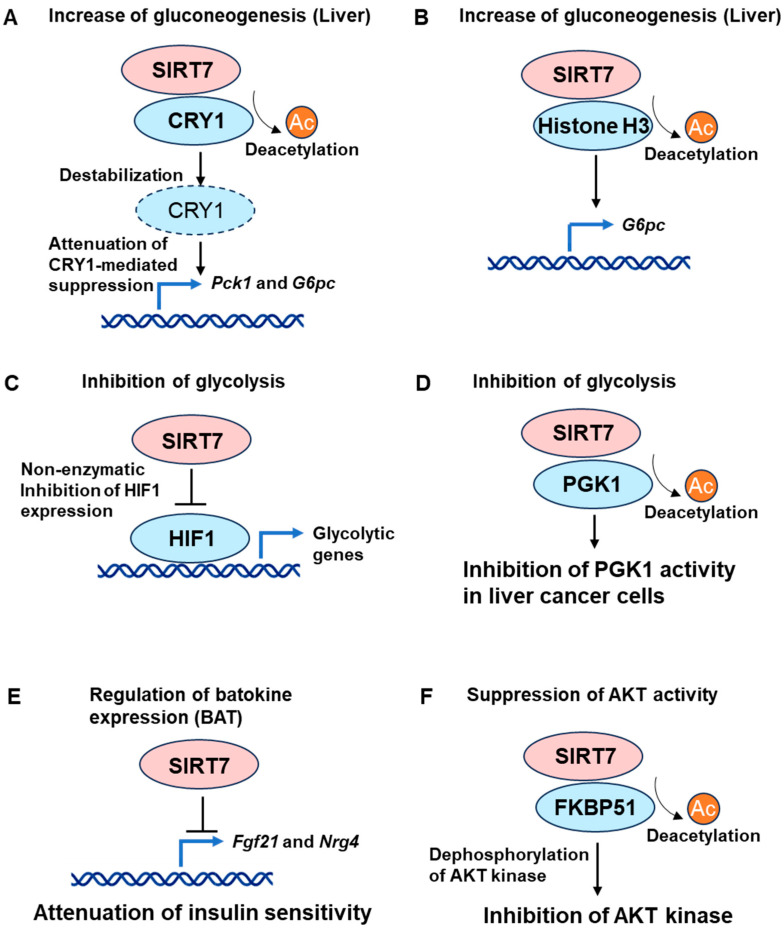



### 3.3. Insulin Action and Glucose Tolerance

Insulin resistance, characterized by an attenuation of the metabolic actions of insulin in the liver, skeletal muscle, and adipose tissue, results from an imbalance between energy intake and expenditure. Obesity is the most common cause of insulin resistance, and endocrine, inflammatory, neural, and cell-intrinsic pathways have been shown to link obesity to insulin resistance [111,112,113,114]. SIRT1 activation by small molecule activators or SIRT1 overexpression improves insulin sensitivity and glucose homeostasis by enhancing fat oxidation in the liver, skeletal muscles, and BAT [69,115,116].

SIRT6 suppresses insulin signaling through the inhibition of key factors in these pathways, including insulin receptor, insulin receptor substrate (IRS)-1, IRS-2, and AKT kinase [117]. SIRT6 deficiency increases insulin-stimulated glucose uptake by activating insulin signaling, which may be involved in severe hypoglycemia in global *Sirt6* KO mice [117]. Consistent with this, pharmacological inhibition of SIRT6 improves glucose tolerance and insulin sensitivity in mice [118]. In contrast, fat-specific and muscle-specific *Sirt6* KO mice exhibit insulin resistance and glucose intolerance [72,81,119]. Transgenic mice overexpressing *Sirt6* are reported to show improved glucose homeostasis [120,121]. Therefore, the regulation of blood glucose levels by SIRT6 is complicated.

*Sirt7* KO mice are resistant to HFD-induced obesity, insulin resistance, and glucose intolerance, indicating that SIRT7 deficiency improves insulin sensitivity and glucose homeostasis in vivo [85]. In contrast, SIRT1 functions to improve insulin sensitivity and glucose homeostasis [69,115,116]. Therefore, SIRT7 and SIRT1 seem to affect insulin sensitivity and glucose tolerance in opposing directions. As stated above, SIRT7 deficiency increases energy expenditure with increased UCP1 expression in BAT [21]. The batokines FGF21 and NRG4 augment insulin sensitivity by increasing energy expenditure in the body [122,123]. Intriguingly, the expression levels of *Fgf21* and *Nrg4* are increased in the BAT of *Sirt7* KO mice (Figure 2E) [21]. Furthermore, hepatic *Fgf21* mRNA expression and the serum FGF21 levels of aged *Sirt7* KO mice are higher than those of control mice [124]. These findings suggest that SIRT7 deficiency improves insulin sensitivity, at least in part, through the enhancement of BAT thermogenesis and overexpression of insulin-sensitizing batokines. Furthermore, SIRT7 suppresses AKT activation by deacetylating FK506-binding protein (FKBP51) in cancer cells under stressed conditions (Figure 2F) [26]. Thus, SIRT7 may directly affect the insulin signaling pathway.

## 4. Conclusions and Future Perspectives

A decade of research has established SIRT7 as an important regulator of glucose and lipid metabolism. As described in this review, SIRT7 regulates numerous metabolic pathways, including adipogenesis, lipogenesis, thermogenesis, and gluconeogenesis, through the deacetylation/deacylation of various target substrates. Intriguingly, the metabolic roles of SIRT7 are quite different from those of SIRT1 and SIRT6 (Figure 3). For example, SIRT1 stimulates lipolysis in WAT in order to stimulate fat oxidation of fatty acids. In contrast, SIRT7 promotes fat synthesis in WAT and the liver. SIRT1 and SIRT6 stimulate thermogenesis in BAT, whereas SIRT7 inhibits thermogenesis. When considering these observations together, it seems that SIRT7 and SIRT1 or SIRT6 regulate metabolism in opposing directions. SIRT1 and SIRT6 play important roles in insulin secretion and control energy balance in the body through modulation of the hypothalamic neurons [125,126,127,128]. It may be interesting to evaluate whether the roles of SIRT7 in pancreatic β-cells and hypothalamic neurons, including proopiomelanocortin (POMC) and agouti-related peptide (AgRP) neurons, are also opposite to those of SIRT1 and SIRT6. SIRT1, SIRT6, and SIRT7 are all located in the nucleus. Another interesting question is how these sirtuins coordinate their actions within cells. Further studies are needed to identify the molecular mechanisms underlying their integration. SIRT1 activation by small-molecule activators improves insulin sensitivity and glucose homeostasis by enhancing fat oxidation in the liver, skeletal muscles, and BAT [69,115,116]. In contrast, SIRT7 deficiency results in resistance to HFD-induced metabolic diseases, including obesity, fatty liver, and glucose intolerance [85]. Therefore, SIRT7 inhibition may be a potential strategy for treating metabolic diseases. There are reports on the development of SIRT7 inhibitors [129,130,131], but further studies are needed to develop more potent and specific SIRT7 inhibitors. In addition, we need to be cautious when considering inhibiting SIRT7 function, given its important role in bone formation and cardiac remodeling [30,43]. Further research will provide new knowledge about the metabolic functions of SIRT7 as well as the effectiveness of SIRT7-selective inhibitors for the treatment of obesity and type 2 diabetes.

## Figures and Tables

**Figure 1 cells-13-00048-f001:**
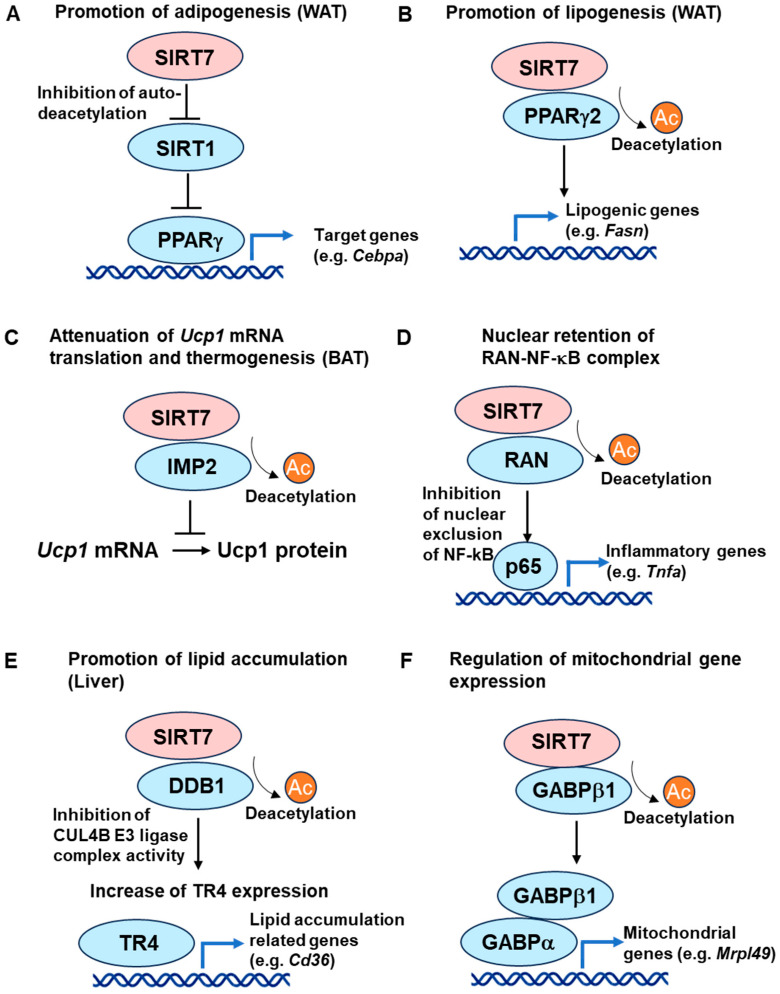
Roles of SIRT7 in lipid metabolism. (**A**) SIRT7 promotes adipogenesis by inhibiting SIRT1 and enhancing the activity of PPARγ [59]. (**B**) SIRT7 also promotes lipogenesis in WAT through the deacetylation of PPARγ2 [20]. (**C**) SIRT7 suppresses thermogenesis in BAT by deacetylating IMP2, an RNA-binding protein that inhibits the translation of *Ucp1* mRNA [21]. (**D**) SIRT7 stimulates inflammation through the nuclear retention of the RAN-NF-κB complex. In the absence of SIRT7, acetylated RAN facilitates the formation of the NF-κB export cargo complex, thereby enhancing the nuclear exclusion of NF-κB [32]. (**E**) SIRT7 deacetylates DDB1, a component of CUL4B/DDB1/DCAF1 E3 ubiquitin ligase complex. SIRT7 increases TR4 protein expression and lipid accumulation in the liver by inhibiting the degradation of TR4 via the suppression of the E3 ubiquitin ligase complex [19]. (**F**) SIRT7 increases mitochondrial gene expression through the deacetylation of GABPβ1 [22].

**Figure 3 cells-13-00048-f003:**
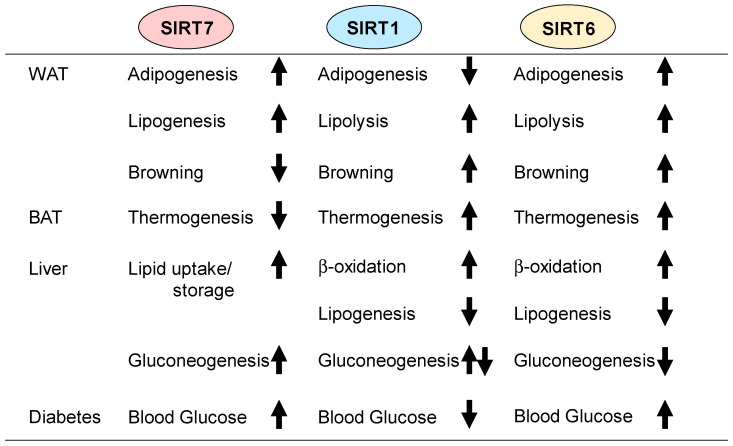
Roles of SIRT1, SIRT6, and SIRT7 in metabolism. Comparison of the metabolic roles of SIRT1, SIRT6, and SIRT7. SIRT1 and SIRT6 stimulate lipolysis in WAT. In contrast, SIRT7 promotes lipogenesis in WAT. In addition, SIRT1 and SIRT6 stimulate thermogenesis in BAT, whereas SIRT7 inhibits it. Although SIRT1 and SIRT6 attenuate lipid accumulation in the liver, SIRT7 increases it. These observations indicate that SIRT7 regulates metabolism in opposite directions from SIRT1 and SIRT6 in some cases.

**Table 1 cells-13-00048-t001:** Target proteins and functions of SIRT7.

Activity		Target Protein	Function
Deacetylation	Gene expression ☐ ☐ ☐ ☐ ☐	Histone H3K18	Tumorigenesis and DNA repair [10,11]
		Histone H3K36/K37	Heterochromatin silencing [12]
		PAF53	Synthesis of pre-rRNA [13]
		U3-55k	Processing of pre-rRNA [14]
		Fibrillarin	rRNA synthesis [15]
		CDK9	RNA polymerase II transcription [16]
	DNA stability ☐	DDX21	Genome stability [17]
		ATM	DNA repair [18]
	Metabolism ☐ ☐ ☐ ☐	DDB1	Lipid metabolism [19]
		PPARγ2	Lipogenesis [20]
		IMP2/IGF2BP2	Thermogenesis [21]
		GABPβ1	Mitochondrial homeostasis [22]
		CRY1	Circadian phase [23]
	Cancer ☐ ☐ ☐ ☐ ☐	WDR77	Transmethylase activity [24]
		PGK1	Glycolysis [25]
		FKBP51	Akt inactivation [26]
		SMAD4	Cancer metastasis [27]
		USP39	Cancer growth [28]
		HAT1	Tumorigenesis [29]
	Cardiorenal disease ☐	GATA4	Regulation of cardiac hypertrophy [30]
		KCC4	Regulation of ion flux [31]
	Immunity	RAN	Regulation of inflammation [32]
	Stress response ☐ ☐ ☐	NPM1	Aging and p53 stability [33]
		FOXO3	Regulation of apoptosis [34]
		p53	Apoptosis [35]
		STRAP	p53 activity and stability [36]
	Stem cell	NRF1	Mitochondrial homeostasis [37]
		NFATc1	Hair follicle initiation [38]
Desuccinylation		Histone H3K122	Chromatin compaction [39]
		PRMT5	Lipid metabolism [40]
Deglutarylation		Histone H4K91	Chromatin structure [41]
Decrotonylation		PHF5A	Aging [42]
Deacylation		Osterix	Bone formation [43]
Mono-ADP ribosylation		SIRT7	Stress response [44]

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
