# Peer review of "The Emerging Role of SIRT7 in Glucose and Lipid Metabolism"

_cells, 2023, doi:10.3390/cells13010048_

Round 1

Reviewer 1 Report

Comments and Suggestions for Authors

The authors provide a concise overview regarding the current understanding of the roles of sirtuins in glucose and lipid metabolism. Here, they focus on SIRT7, given that the cellular and biological role of SIRT7 has only been uncovered recently in more detail (most of the 511 SIRT7-related papers have been published between 2015 and Dec. 2023; Pubmed search performed on Dec. 5th, 2023).

Overall, the review is concise, well structured, and benefits from the overview Figures 1 to 3. Figure 3 summarizes the roles of SIRT7 as compared to the also nuclear functional SIRT1 and SIRT6 in various aspects of metabolic regulation. However, Table 1 summarizing the currently known SIRT7 targets needs some re-organisation. In terms of deacetylation, the target proteins and their functions should be sub-grouped/better sorted to provide an easier overview fort he reader.

The authors mention the possibility of SIRT7 inhibition in the treatment of metabolic diseases. This is likely challenging. Are there any strategies for the development of SIRT7-specific inhibitors available? What is the prognosis of the authors in this regard?

Lastly, there are few typos in the manuscript (e.g., Table 1. SIRT7 targets).

Author Response

Response to the comments of Reviewer 1

We are grateful for the reviewer’s comment that “the review is concise, well structured, and benefits from the overview Figures 1 to 3.” and for the constructive suggestions. We have revised the manuscript based on these valuable comments.

However, Table 1 summarizing the currently known SIRT7 targets needs some re-organisation. In terms of deacetylation, the target proteins and their functions should be sub-grouped/better sorted to provide an easier overview for the reader.

We thank the reviewer for pointing this out. Based on the reviewer’s suggestion, we have reorganized Table 1. In the new Table 1, SIRT7 target proteins were sub-grouped according to their biological roles (i.e., gene expression, DNA stability, metabolism, cancer, cardiorenal disease, immunity, stress response, and stem cell).

The authors mention the possibility of SIRT7 inhibition in the treatment of metabolic diseases. This is likely challenging. Are there any strategies for the development of SIRT7-specific inhibitors available? What is the prognosis of the authors in this regard?

At present, there are no commercially available SIRT7 inhibitors. Although there are several reports on SIRT7 inhibitors, their specificity seems to be insufficient. Based on the reviewer’s suggestion, we have added the following sentence to the revised manuscript: “There are reports on the development of SIRT7 inhibitors [129-131], but further studies are needed to develop more potent and specific SIRT7 inhibitors.” (Conclusions and future perspectives; lines 356 to 357).

Lastly, there are few typos in the manuscript (e.g., Table 1. SIRT7 targets).

We apologize for the errors in the original manuscript. We have changed the title of Table 1 to “Target proteins and functions of SIRT7” and had the manuscript checked by an English editing service.

Reviewer 2 Report

Comments and Suggestions for Authors

The review on the emerging role of SIRT7 is well-written and Figure 3 and Table 1 are very informative, though the information from Table 1 is not in the text. I suggest briefly referring to other roles of SIRT7 that are not described in this manuscript in the introductory section and providing references to recent reviews on the topics.

Figures 1 and 2 appear fragmented; the authors may consider putting SIRT7 in the centre and depicting these roles/interactions/pathways around it.

Stress responses are mentioned in the abstract, but there is no description of the SIR7 role in these processes. Referring interested readers to the stress response roles and especially emphasizing the role of SIRT7 in metabolic stress response and ageing may increase the visibility of this review (Raza U, Tang X, Liu Z, Liu B. SIRT7: the seventh key to unlocking the mystery of ageing. Physiol Rev. 2024 Jan 1;104(1):253-280).

Author Response

Response to the comments of Reviewer 2

We are grateful for the reviewer’s comment that “The review on the emerging role of SIRT7 is well-written and Figure 3 and Table 1 are very informative” and for the constructive suggestions. We have revised the manuscript based on these valuable comments.

The review on the emerging role of SIRT7 is well-written and Figure 3 and Table 1 are very informative, though the information from Table 1 is not in the text. I suggest briefly referring to other roles of SIRT7 that are not described in this manuscript in the introductory section and providing references to recent reviews on the topics.

We thank the reviewer for pointing this out. We have revised the Introduction section with the addition of the following underlined text: “Since this breakthrough, a number of target proteins and physiological functions of SIRT7 have been identified (Table 1) [10-44] and it is now clear that SIRT7 plays critical roles in many biological processes, including gene expression, genome stability, stress response, and metabolism.” (lines 39 to 41 in the revised manuscript), and “Other cellular and biological roles of SIRT7, such as stress response, are covered in several recent reviews [48-51].” (lines 47 to 48).

Figures 1 and 2 appear fragmented; the authors may consider putting SIRT7 in the centre and depicting these roles/interactions/pathways around it

In accordance with the reviewer’s suggestion, we have added more information about SIRT7, including its roles, interactions, and pathways, to the Figures. We thank the reviewer for the suggestion to put SIRT7 in the center and display information around it. However, we decided to show the metabolic roles of SIRT7 separately in order to provide an easier-to-understand overview. Thank you for your understanding.

Stress responses are mentioned in the abstract, but there is no description of the SIR7 role in these processes. Referring interested readers to the stress response roles and especially emphasizing the role of SIRT7 in metabolic stress response and ageing may increase the visibility of this review (Raza U, Tang X, Liu Z, Liu B. SIRT7: the seventh key to unlocking the mystery of ageing. Physiol Rev. 2024 Jan 1;104(1):253-280).

We thank the reviewer for pointing this out. We have newly cited the recommended review and added it to the References (reference 51: Raza U. Physiol Rev 2024). We have added the following sentence to the revised manuscript: “Other cellular and biological roles of SIRT7, such as stress response, are covered in several recent reviews [48-51].” (lines 47 to 48). 
